# Multifunctional Portable System Based on Digital Images for In-Situ Detecting of Environmental and Food Samples

**DOI:** 10.3390/molecules28062465

**Published:** 2023-03-08

**Authors:** Diego Barzallo, Jorge Benavides, Víctor Cerdà, Edwin Palacio

**Affiliations:** 1Environmental Analytical Chemistry Group, Department of Chemistry, University of the Balearic Islands, 07122 Palma, Spain; 2Department of Electrical and Electronic Engineering, Universidad del Valle, Cali 760042, Colombia; 3Sciware Systems, 07193 Bunyola, Spain

**Keywords:** portable device, 3D printed, matlab, webcam, digital-image-based colorimetry, fluorimetry

## Abstract

The development of a portable device created by 3D printing for colorimetric and fluorometric measurements is an efficient tool for analytical applications in situ or in the laboratory presenting a wide field of applications in the environmental and food field. This device uses a light-emitting diode (LED) as radiation source and a webcam as a detector. Digital images obtained by the interaction between the radiation source and the sample were analyzed using a programming language developed in Matlab (Mathworks Inc., Natick, MA, USA), which builds the calibration curves in real-time using the RGB colour model. In addition, the entire system is connected to a notebook which serves as an LED and detector power supply without the need for any additional power source. The proposed device was used for the determination in situ of norfloxacin, allura red, and quinine in water and beverages samples, respectively. For the validation of the developed system, the results obtained were compared with a conventional spectrophotometer and spectrofluorometer respectively with a *t*-test at a 95% confidence level, which provides satisfactory precision and accuracy values.

## 1. Introduction

According to previous research [1,2,3,4], current trends in analytical monitoring of environmental and food samples include the following: miniaturization of used equipment; the potential of in situ analysis; development of rapid response spot test methods; the decrease in the amount of sample and the reduction in the amount of organic solvents [5]. Today, spectroscopic methods are commonly used for quantitative chemical analysis through the use of many instruments such as spectrophotometers [6], colorimeters [7], spectrofluorometers [8], which measure the amount of light absorbed or emitted by an analyte, based on the concentration present in the sample at specific wavelengths.

In this sense, light sources are an indispensable component of a large number of analytical methods. The development of new and miniaturized light sources in comparison with unlike traditional light sources such as tungsten bulbs, has allowed the advance of inexpensive, miniaturized, and robust methods. An example of this is the LED, which has been coupled into portable microfluidic chips for fluorescent detection [9,10] or integrated both into a multi-syringe flow injection analysis (MSFIA) fluorimetric system [11] or a lab-in-syringe flow system exploiting dispersive liquid-liquid micro-extraction for spectrophotometric detection [12]. In addition, several detectors [13] based on digital images have allowed the increasing use of in-situ spectroscopic techniques [14,15]. Specifically, absorbance and fluorescence measurements have found new applications in the environmental and food field through the use of portable equipment based on 3D printing, which are intended to be an alternative to commercial analytical instruments [2].

3D printing (additive manufacturing) allows the design and manufacture of multiple prototypes of different geometry and shape, which are usually manufactured using extrusion techniques (e.g., fused deposition modelling) or photopolymerization (stereolithography) according to the type of application [16,17]. In addition, smartphones have been commonly used as detectors in these miniaturized analytical devices for spectrophotometric [18,19], chemiluminescent [20] or fluorometric [21] measurements. Other proposed systems have used digital cameras [22,23]. However, there are only a few systems that have used a webcam for spectrophotometric measurements [24,25] and especially for fluorometric measurements [26]. In any case, they require subsequent processing of the digital images obtained through the use of different programs such as: ChemoStat, Photoshop Adobe, Image J, which use image decomposition models such as HSV [27], CIE L *a* b * [28] or RGB color [29] for this purpose. After this, an Excel sheet should be used for the treatment of the quantitative data obtained (e.g., construction calibration curves) and subsequent acquisition of results [14,30,31]. This implies a greater intervention of the analyst, certain knowledge in the use and handling of the image processing program, and greater analysis times of unknown samples. Considering this background, a programming language has been developed using the Matlab software that allows the decomposition of the images obtained in real-time in a single step, using the RGB model (red, green and blue).

On the other hand, fluoroquinolone antibiotics (FQs) such as norfloxacin (NOR) are widely used in human medicine to treat diseases of the urinary tract [32]. These are usually poorly absorbed by the human body and excreted in their original form or transformed through urine and faeces, which have been detected in various compartments of the environment, e.g., wastewater [33], reclaimed water [34], surface water [35] and even drinking water [36]. Solid phase extraction (SPE) has been commonly used as sample pretreatment to evaluate these antibiotics due to the low concentrations found in environmental samples [32] and subsequent quantification by high performance liquid chromatography with mass spectrometry (HPLC-MS) [37], due to its greater selectivity compared to other detectors. As a proof of concept, SPE has been combined with colorimetric detection for its analytical determination using the developed system. Food additives such as allura red and quinine have been also commonly used and consumed in the food industry. Allura red (AR) is a synthetic azo dye that provides odour, colour, aroma, and flavour to products such as sauces, drinks, candy, and cereals [38]. However, its excessive consumption causes consumer health problems including nausea, asthma, allergic reactions, and cancer [39]. For this reason, the European Food Safety Authority (EFSA) established that the maximum concentration of Allura Red in drinks must be 100 mg L^−1^ [40]. In addition, the main use of the quinine is as flavouring for tonic drinks due to its bitter taste. The European Union legislation established a maximum limit of Quinine in soft drinks of 100 mg L^−1^ due to it causing health problems by exceeding the permissible limit [11,41]; so it is essential to determine the concentrations of both food additives in soft drinks.

Therefore, the main objective of this work is to combine the development of a miniaturized system based on 3D printing using a webcam as a detector and subsequent quantitative determination of coloured solutions from the RGB values obtained from the processing of digital images using software developed in Matlab. There are no previous studies of miniaturized devices that combine absorbance and fluorescence measurements simultaneously with some software that includes all the necessary components and algorithms to perform both analytical measurements. In addition, the idea behind this is to involve Analytical Chemistry laboratory students in the use of 3D printing and consolidate the theoretical foundations such as Beer’s law for the quantitative determination of coloured solutions with very friendly software that does not require image experience. The performance of the developed system was demonstrated for the in-situ analysis both of norfloxacin in environmental samples and allura red/quinine in food samples.

## 2. Results and Discussions

### 2.1. Optimization of Parameters of the Proposed System

A univariate method was used to optimize the parameters that affect measurements performed with the proposed system, including the type of cuvette, the distance of the webcam with respect to the cuvette, and LED current intensity. Three types of cuvette: quartz, plastic, and glass were evaluated with respect to the fluorescence intensity. The best results were reported to use a quartz cuvette which provides increased sensitivity to the instrument due to the quality of the material and the ability to not absorb radiation in the range being worked on compared to other cuvettes used (Figure 1A) [42]. Therefore, it was used for further colorimetric and fluorimetric measurements. In this way, the influence of the webcam distance with respect to the cuvette has been studied. Figure 1B shows the best results when locating the webcam at 2 cm from the cuvette, which allowed to ensure an optimal focus. However, locating the detector at further distances means increasing the digital zoom of the webcam to crop the image at that point, which reduces sharpness and image quality according to the obtained results [26]. In addition, to keep the webcam static at this distance, a lateral hole was made in the 3D support where a small tube was inserted to adjust it, which provides better reproducibility in further experiments. Finally, the influence of the radiation current intensity was evaluated. In this way, for colorimetric measurements carried out with norfloxacin and allura red solutions, no differences were observed in the absorbance values obtained when applying different current intensities (Appendix A), since they are not proportional to the intensity of the radiation current used, but rather to its concentration according to the Lambert–Beer law [43]. However, for fluorimetric measurements the color of the detected image obtained from quinine standard solutions using middle and high (5V) current intensity is increasing with the current intensity of the radiation source (Appendix A). Pokrzywnicka et al. [44] verified that by increasing the intensity of the light-emitting source with the use of two LEDs, its resolution increases, and improves the sensitivity of the device. In addition, the images obtained are rectangular shape because they were cropped by Matlab to minimize the effect of possible light scattering from the internal walls of the 3D holder and the walls of the cuvettes.

### 2.2. Method Quality Assurance

Norfloxacin determination was based on a previous work performed by Lamarca et.al. [14] with some modifications that include solid phase extraction (SPE) [45]. Satisfactory norfloxacin recovery percentages were obtained with a standard solution of 0.5 mg L^−1^ at neutral pH, which was similar with reported results by Ngumba et al. [46] in the pH range 4–7. Thus, it was chosen for further measurements due to its in-situ analysis. In addition, 100 mL of sample was chosen for pre-concentration procedure to obtain a high sensitivity, however, higher sample volumes were not studied, because it is carried out in situ and manually. Figure 2A. shows absorption spectra obtained from the norfloxacin reaction with chloranilic acid using a spectrophotometer. The product formed showed a purple color immediately after the reaction was performed, whose intensity increases with the concentration of NOR. Colorimetric measurements using the 3D device were made with a 500 nm monochromatic LED to selectively excite the sample (increase its sensitivity), which also avoids the use of filters or monochromators [11], then the digital images were captured and processed through the software developed in Matlab. The best results were obtained with a blue calibration curve as shown in Figure 2B.

Allura red determination was based on the method proposed by FAO [39,47] and the quinine method was based on the study proposed by Da Silva et al. [15]. Likewise, pH values of allura red and quinine standard solutions were previously optimized with a standard solution of 0.5 and 3 mg L^−1^ respectively, which were adjusted in the pH range 2–8. The best results with allura red solutions were obtained at neutral pH, so it was not necessary to adjust the solution pH for subsequent measurements, similar to the study reported by Rovina et.al [39]. 

Figure 2C, shows absorption spectra obtained of allura red standard solutions (0.1–0.5 mg L^−1^) using a spectrophotometer. Measurements of colorimetric were performed with the same monochromatic LED used above. As can be seen in Figure 2D, very satisfactory results were obtained with the blue and green colors, from the complementary red color (allura red solution), which were similar to those obtained by Danchana et al. [40], where they used a 510 nm green LED and obtained the best results in the blue color.

In addition, the variation in fluorescence intensity of quinine solutions increased with the decrease in the solution pH [15,48]. Therefore, pH 2 was chosen as the working pH for further works. Figure 2E shows absorption spectra obtained of quinine solutions standards (0–0.5 mg L^−1^) using a spectrofluorometer. RGB calibration curves were obtained using a monochromatic UV light of 351 nm as the excitation source. The best results were reported in the blue color due to quinine is colorless when it is excited by the radiation source, and emits a blue light beam (440–490 nm) [49]. In addition, green and red color values cannot be used in these determinations, due to they did not show wide linearity and provided poor sensibility as shown in Figure 2F.

### 2.3. Analytical Parameters

Table 1 includes the results obtained with the proposed system which were compared with a conventional optical instrument. For all determinations, the linear range of study was performed by triplicate (n = 3) and the means of the values were obtained as analytical responses, which were the same compared to each reference method. In addition, for the proposed system there was no loss of linearity with a correlation coefficient (r^2^) ≥ 0.993, which shows a great advantage for its applicability. The limits of detection (LOD) and quantification (LOQ) were calculated as the concentration corresponding to three times and ten times the standard deviation of the blank solution (n = 10) over the calibration slope, respectively [50]. The precision of the proposed system was evaluated in terms of repeatability (intra-day) and reproducibility (inter-day) expressed as relative standard deviation (%RSD) by analyzing triplicate samples. The values obtained varied between 0.6 and 3.7% and between 1.5 and 9.6% for intra-interday, respectively. Moreover, Appendix A show the execution of the Matlab software R2021a performing absorbance and fluorescence measurements respectively.

In addition, the preconcentration factor was calculated in the proposed method for norfloxacin determination. This was obtained as the ratio between the slope of the regression line performing the preconcentration procedure and that obtained without preconcentration, that is 31 with 100 mL of sample. Furthermore, the extraction efficiency using the developed 3D device and the spectrophotometer were close to 84%. Meanwhile, without enrichment the LOD obtained using the proposed device was only 1.4 mg L^−1^ and the LOQ was 5 mg L^−1^ and with enrichment were 0.05 and 0.16 mg L^−1^, respectively.

### 2.4. Application and Validation

Environmental and food samples were analyzed by digital imagen-based method and spectrophotometry, whose results were compared with a *t*-test at a 95% confidence level as shown in Table 2. Based on the results obtained in the determination of allura red in beverages samples, the highest *t*-test value was 2.2 which is lower than the critical *t*-test value of 2.91 for n = 3. Thus, there is no significant statistical difference in the concentrations obtained with both systems. Besides, these concentrations are below the limit reported by EFSA, that is 100 mg L^−1^, reported similar results in other works [51,52,53,54]. Possible interferences of traces of heavy metals, cations, anions, flavorings, or colorants used in carbonated drinks can affect the quantification of allura red such as the iron cation that has a maximum absorption at 510 nm. To assess matrix effects, all samples were analyzed without and with spikes of 1 mg L^−1^. Good recoveries were obtained in the range of 101–104%, which indicates no significant matrix effects in quantitative analysis. These results obtained were corroborated with those reported by Bişgin et.al [38], who found that there is no interference effect in the determination of allura red in beverage samples.

Similarly, tonic beverage samples containing quinine were analyzed with the proposed system, and spectrofluorometer as shown in Table 3, reported the highest *t*-test value of 2.0 which is lower than the critical *t*-test value of 2.91 for n = 3, which means that concentrations obtained are similar with both systems. In addition, the results obtained comply with the provisions of European Union legislation, which indicates that the maximum concentration of quinine must be 100 mg L^−1^. These results were similar to those obtained by other authors using other analytical techniques [55,56,57]. In addition, in a previous work carried out by authors [11], the interference effects in quinine determination from beverage samples were reported, where the only interferent that can quench the quinine fluorescence is chloride ion. However, it does not interfere at concentrations below 0.4 mM, which generally is the content in a tonic drink [58,59].

### 2.5. Comparison with Other Previous Methodologies

Currently, there are several methods based on digital images using low-cost detectors. Oliveira et al. [60] for the determination of Cu^2+^ in distilled beverages used a smartphone as a detector and the image processing to obtain RGB data was performed using a Color Grab mobile application. Danchana et al. [43] for colorimetric determination of iron (II) and hypochlorite in water used a webcam as a detector. Images were captured using the YouCam program and the deconvolution of the colors was carried out by ImageJ to obtain RGB data. In both cases, the RGB values obtained with any application require a further treatment for the interpretation of the results, i.e., conversion of RGB data to absorbance values, construction of calibration curves, and linear regression fit.

The novelty of this 3D printed system with respect to those mentioned above and other colorimetric systems that have also been reported in the literature [14,49,61], is the versatility to perform both colorimetric (Appendix A) and fluorimetric (Appendix A) measurements simultaneously from the processing of digital images in real-time through the software developed in Matlab that includes various advantages: (i) choice of the analytical technique to be used, (ii) capture of the digital images, (iii) construction of the calibration curves and (iv) rapid acquisition of RGB channel results. 

Other benefits are that it requires less analysis time, no experience in the treatment of digital images and it has been applied as a proof of concept to environmental and food samples, which presents greater precision and better sensitivity compared to other systems reported in the literature as shown in Table 4. In addition, it provides an excellent screening method for in-situ detection of total fluoroquinolones in water samples before using more sensitive and more expensive reference analytical methods.

## 3. Methods and Materials

### 3.1. Chemicals and Samples

All reagents and solvents were of analytical reagent grade. For colorimetric measurements, a stock solution of 500 mg L^−1^ was prepared by dissolving 0.050 g of allura red (Sigma-Aldrich, Darmstadt, Germany) in 100 mL deionized water. Working standard solutions were daily prepared by stepwise dilution of AR stock solution with distilled water. A stock solution of 500 mg L^−1^ of norfloxacin (NOR) (Sigma-Aldrich, Darmstadt, Germany) was prepared in acetonitrile. Working standard solutions were made by an appropriate dilution of NOR stock solution with distilled water and 0.1% (*m*/*m*) chloranilic acid (CL) prepared in acetonitrile. Samples’ pH were adjusted with 0.5 mol L^−1^ of H_2_SO_4_ and NaOH, respectively. Solid phase extraction (SPE) of NOR was carried out using Oasis HLB resin (200 mg, 30 µm), which allows the retention of hydrophilic and lipophilic compounds.

For fluorometric measurements, a stock solution of 500 mg L^−1^ was prepared by dissolving an adequate amount of quinine sulphate dihydrate (Scharlau, Barcelona, Spain) in 0.05 mol L^−1^ sulfuric acid (Scharlau, Barcelona, Spain). Working standard solutions were made by an appropriate dilution of the quinine stock solution in 0.05 mol L^−1^ sulfuric acid. In addition, all stock solutions prepared were stored under refrigeration at 4 °C for a month in an amber bottle to avoid light radiation.

### 3.2. Environmental and Food Samples

Treated wastewater sample was obtained from a wastewater treatment plant (WWTP) (Palma de Mallorca, Spain), and commercialized mineral water was purchased from a local market. All samples were filtered through a cellulose membrane filter of pore size 0.45 um (Millipore Ibérica, Madrid, Spain).

Beverage samples were purchased from a local supermarket (Palma de Mallorca, Spain). All samples were previously degassed using an ultrasonic bath for five minutes and stored at room temperature. Subsequently, allura red and quinine samples were diluted in a volumetric flask with deionized water and 0.05 mol L^−1^ sulfuric acid solution, respectively.

### 3.3. Column Preparation for Environmental Samples

A plastic column (Mobicol, Emsdetten, Germany) with dimensions of 2 cm × 8 mm i.d. was packed following the procedure described by Vargas-Muñoz et al. [64] with some modifications. 500 mg of Oasis HLB resin was packed and two glass wool frits were placed at the ends of the column to avoid loss of SPE resin. This column was connected to a variable volume syringe to perform the aspiration or dispensing of samples and solvents to carry out the preconcentration of NOR. In addition, the same column was used in all experiments.

### 3.4. Preconcentration Procedure and Colorimetric Analysis

The SPE sorbent was previously conditioned using 5 mL of methanol and 5 mL of water at neutral pH. Samples were loaded onto the column. Afterwards, the column was rinsed with distilled water to remove some interferents and norfloxacin was eluted with methanol. After each elution, the column was washed twice with 5 mL of distilled water. The eluates were evaporated using a portable USB heater at 65 °C (Runmeihe, Amazon.es, Spain) and redissolved in 2.7 mL of acetonitrile (100%) with 300 µL of chloranilic acid into a cuvette for further colorimetric analysis based on digital images as shown in Figure 3.

## 4. System Description

Figure 4, shows a real image of the proposed system developed with a detailed identification of each component. It proposed system consists of a 3D holder which houses a light-emitting diode (LED) as a radiation source, a cheap webcam for the detection of the analytical signal, an electronic circuitry (located inside the 3D part) which allows controlling the intensity of the light source, a head based on 3D printing which houses the quartz cuvette, a USB tester (optional) which directly measures the amount of light intensity entering the system and the software Matlab R2021a (Mathworks Inc., Natick, MA, USA), which builds the calibration curves at real-time using a notebook. The entire system is powered through the USB outputs of this laptop without the need for any additional power source. A more detailed description of the components is shown below.

### 4.1. Components

#### 4.1.1. 3D Holder

3D holder was designed using the Rhinoceros program taking into account the dimensions and geometry of each component that makes the system up: webcam, LED, and cuvette. Appendix A includes the design 3D holder which was fabricated with black methacrylate resin using FormLabs2+ stereolithographic printer. It has a square geometry hole (size 14 × 14 × 45 mm) to locate the cuvette, which is surrounded by rectangular openings where mirrors have been inserted to prevent the dispersion of the light beam. Circular holes (6.2 mm) with respect to the cuvette were designed to locate the 5 mm LEDs according to chemical analysis. In addition, a suitable head (size 30 × 30 × 16.4 mm) that covers the cuvette has been designed to avoid outside light entered into the system which could cause irreversibility and incorrectness in the results obtained.

#### 4.1.2. Radiation Source

In both methods, different types of conventional monochromatic LEDs were used as excitation source, according to the type of analyte to be determined, since it improves selectivity and simplifies signal interpretation [43]. In addition, the excitation spectrum of the monochromatic LED used for fluorescent measurements was monitored using a small USB2000 CCD detector (Ocean Optics Inc., Dunedin, FL, USA) with SpectraSuite 5.1 software, where a maximum absorption signal was obtained at 351 nm, while not any additional analytical signal was obtained in other regions of the spectrum. Thus, this provides great selectivity to the proposed system, which was corroborated with the spectrum obtained by Da Silva et al. [15] when a monochromatic LED at this same wavelength was used for the quantification of quinine in beverages.

LED was placed in line with the detector to collect the light transmitted by the sample for colorimetric measurements as a conventional spectrophotometer. For fluorometric measurements, it was placed on the side of the 3D holder forming an angle of 90° with respect to the detector to avoid reflected or transmitted incident light reaching the detector as a conventional spectrofluorometer. Furthermore, it was placed at an adequate distance from the cuvette to ensure that the sample absorbs this excitation light and emits an almost directed light beam which ensures that the results obtained based on the color intensity of the images were correct. In addition, an electronic circuitry directly connected to a USB of a notebook has been built to control the intensity of the light emitted by a LED as shown in Appendix A.

#### 4.1.3. Detector

A webcam (Microsoft LifeCam Cinema, Amazon.es, Spain) has been selected as a detector due to its good resolution, low cost, and portability (Appendix A). This device has a 5 Mpx camera (autofocus), a CMOS sensor, and a high-precision glass lens, which allow the capture of high-quality digital images even in low light conditions (photonic noise reduction) [38]. In addition, the color captured by the CMOS sensor of the webcam records the primary colors: red (R), green (G), and blue (B) that produce the digital image. In this way, the color variation of the digital images obtained can vary from 0–255 (8-bit format), according to the criteria established by the International Color Consortium (CIC) [49].

### 4.2. Matlab Program

An executable interface was developed in MATLAB for the quantitative analysis of samples from digital images using absorbance or fluorimetric measurements. A more detailed description of the program’s functions is detailed below.

#### 4.2.1. Programming Language Development

A series of instructions and operations were programmed in Matlab software for image processing and quantitative data acquisition through the RGB model as shown in Appendix A. The main window includes: (A) Command Window, which indicates the instructions to execute; (B) Current Folder and Workspace, which contains the saved files and the variables/objects used, respectively; (C) Menu bar with Run option to execute the developed algorithm. It was coupled to a graphical user interface development environment (Guide) that executes a visual and graphical window, which allows real-time capture, processing, and analysis of the digital images obtained from each sample contained inside the cuvette of the proposed system. In addition, this program provides easy handling and shorter analysis time compared to other digital image-based systems [22,43].

#### 4.2.2. Quantitative Analysis of Digital Images

For absorbance measurements, the program developed converts the RGB color data obtained from image processing into absorbance values by the ratio A= log I_0_/I_s_, where I_0_ is the blank intensity value and I_s_ is the intensity value of each standard or sample expressed in both cases for each RGB color. For fluorescence measurements, the obtained RGB color values were plotted directly on the calibration curve as intensity values versus the concentration of each standard or sample. As can be seen, the webcam coupled to the developed program acts as a multifunctional optical detector for absorbance and fluorescence measurements in a single device compared to conventional optical equipment.

#### 4.2.3. Digital Image Processing

Figure 5, shows the executable MATLAB program developed for the treatment of the images captured by the detector.

The first segment (A) corresponds to the capture of digital images by webcam in real time where the analyst chooses which analytical technique will use for the quantification of the analyzed samples. The intensity values of each RGB channel were obtained from the cropped, square, and homogeneous digital images made by the program, which also allows the capture of 60 images in one minute of every sample and provides an average value of each color RGB. In addition, the program indicates the analyst to carry out the measurements of the blank and subsequently of the standards or samples. 

The second segment (B) allows the construction of the calibration curves for each color: red, green, and blue, which indicates the relationship between standard solution concentration and the color intensity. In addition, it contains a numerical table that includes the concentration of standards and intensity values obtained from image processing. 

Segment (C) provides the results obtained from the quantitative analysis after the image processing has finished. It provides both the correlation coefficient and the linear equation of each calibration curve. The best calibration curve was chosen to take into account the higher slope and better linearity from the three RGB channels which afford to determine the analyte content of the sample.

### 4.3. Spectrophotometer and Spectrofluorometer for Comparative Purposes

For comparison purposes, a conventional spectrophotometer (LS-50DFI, Hewlett Packard, Palo Alto, CA, USA) was used for the determination of norfloxacin and allura red with wavelengths of maximum absorption at 500 nm. Quinine quantification was developed in a spectrofluorometer (LS-50B., Perkin Elmer^®^ Inc., Waltham, MA, USA) with excitation and emission wavelengths at 351 nm and 447 nm, respectively.

## 5. Conclusions

A miniaturized system based on 3D printing was developed using a web camera as a detector for colorimetric and fluorometric measurements. The applicability of this system was demonstrated in the determination of norfloxacin in environmental samples; allura red and quinine in beverage samples, obtaining results very similar to those obtained with a conventional optical equipment, through statistical analysis with a *t*-test at 95% confidence level. In addition, the precision and accuracy results obtained with the developed system are satisfactory due to the optimization carried out of each system parameter and the use of the developed software in Matlab which allows the construction of the calibration curves directly, minimizing the intervention of the analyst in images processing being a great advantage compared to other manual programs. Due to its low cost and portability, it offers a great alternative for in situ or laboratory analysis of other samples in the food or environmental field and as a result of its miniaturized size, it provides a minimization of the waste generated by samples and instrumentation causing a positive effect on the environment.

In this way, it is intended to resolve the limitations of photometry and fluorimetry in various fields of analysis related to instrumentation and miniaturization problems. Thus, it also is a good practical exercise in student learning that combines the use of 3D printing with friendly digital image quantification software to consolidate theoretical knowledge of spectroscopy.

## Figures and Tables

**Figure 1 molecules-28-02465-f001:**
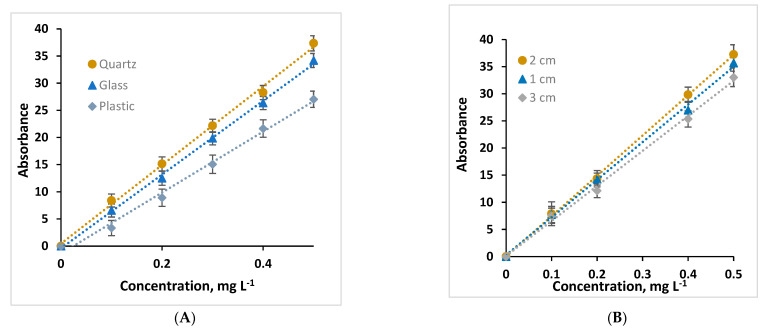
(**A**) Analysis of the type of cuvette to be used, (**B**) Influence of the webcam distance from the cuvette. Working conditions: 0–0.5 mg L^−1^ quinine sulphate diluted in 0.05 M acid sulfuric.

**Figure 2 molecules-28-02465-f002:**
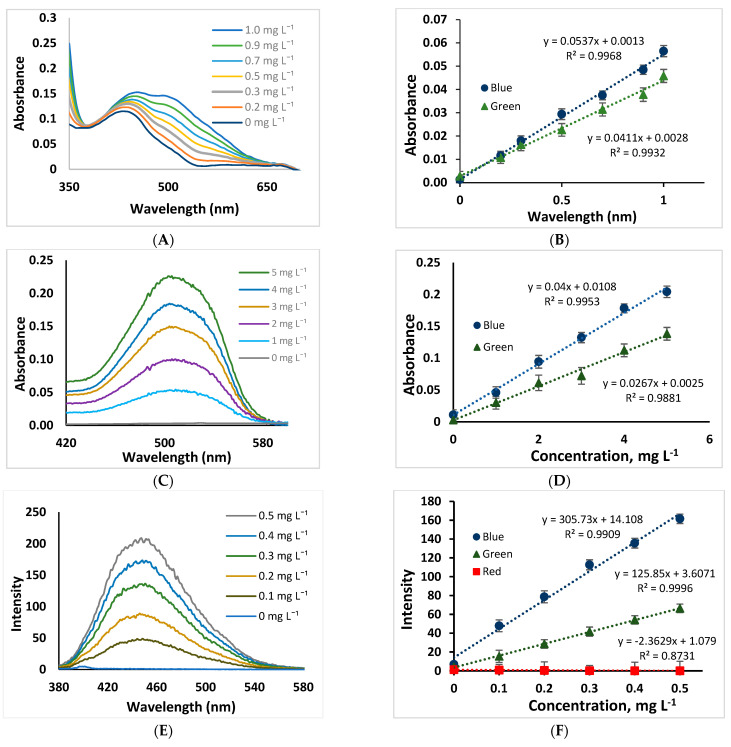
Colorimetric measurements: Absorption spectra obtained with: (**A**) after preconcentration procedure from norfloxacin solutions and (**C**) allura red solutions. Calibration curves obtained using the proposed system: (**B**) norfloxacin solutions and (**D**) allura red solutions. Fluorimetric measurements: (**E**) emission spectra of quinine solutions and (**F**) calibration curve obtained using the proposed system. Error bars represent standard deviation (*n* = 10).

**Figure 3 molecules-28-02465-f003:**
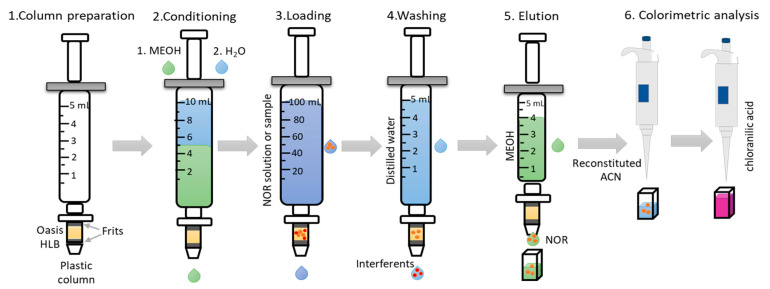
Schematic representation of the norfloxacin preconcentration procedure for further colorimetric analysis based on digital images.

**Figure 4 molecules-28-02465-f004:**
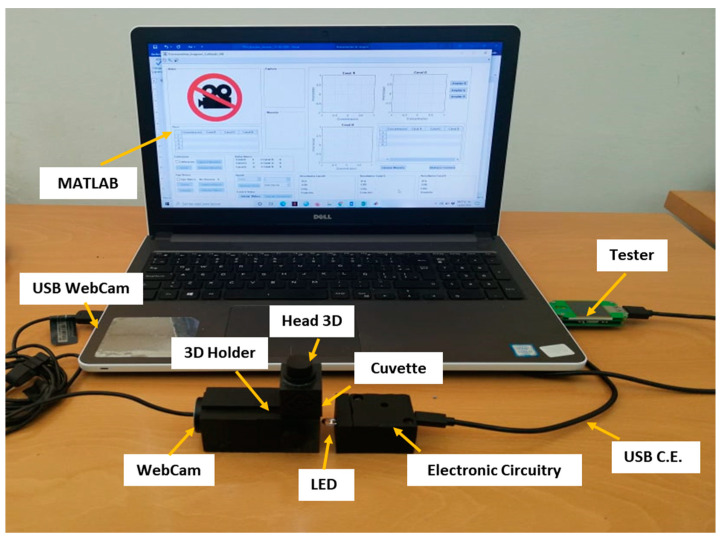
Real proposed system used for colorimetric and fluorometric measurements.

**Figure 5 molecules-28-02465-f005:**
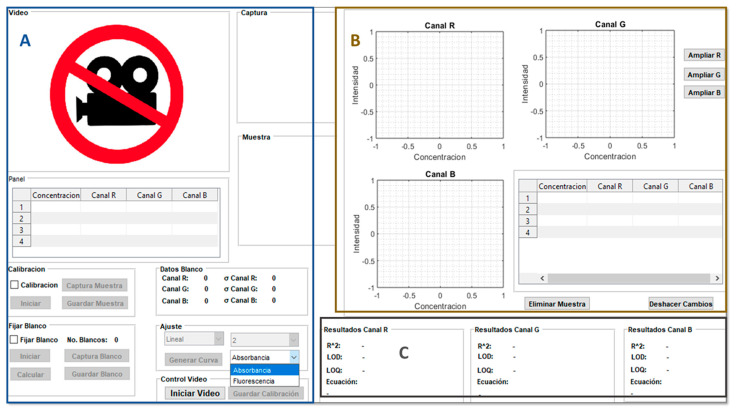
Digital image processing by program Matlab developed.

**Table 1 molecules-28-02465-t001:** Comparison of the results obtained using the proposed system with each conventional optical instrument.

Analytical Parameters	NOR Determination		AR Determination		Quinine Determination	
	Webcam * 500 nm	Spectrophotometer	Webcam * 500 nm	Spectrophotometer	Webcam ** 351 nm	Spectrofluorometer
Calibration curve (n = 3)	A = 0.0537 [NOR] + 0.0013	A = 0.997 [NOR] + 0.0431	A = 0.040 + [AR] + 0.0108	A = 0.0451 [AR] + 0.0028	I = 305.73 [quinine] + 14.108	I = 412.86 [quinine] + 4.565
Correlation coefficient (r^2^)	0.995	0.995	0.993	0.998	0.996	0.997
Linearity range (mg L^−1^)	0.2–1.0	0.2–1.0	1–5	1–5	0.1–0.5	0.1–0.5
LOD (mg L^−1^)	0.06	0.04	0.25	0.23	0.02	0.001
LOQ (mg L^−1^)	0.20	0.15	0.85	0.75	0.06	0.005
Repeatability (%RSD) (n = 5)	1.4–3.7	1.9–3.3	0.7–1.3	0.7–1.0	1.3–2.6	0.6–1.7
Reproducibility (%RSD) (n = 5)	2.7–9.6	2.4–5.7	2.6–3.4	1.5–2.5	2.8–4.1	1.5–2.6

*: Middle current intensity of the LED; **: High current intensity of the LED.

**Table 2 molecules-28-02465-t002:** Concentrations of norfloxacin and allura red obtained from colorimetric measurements in environmental and food samples (n = 3).

Samples	Webcam System				Spectrophotometer	*t*-Test
	Dilution Factor ^c^	Added mg L^−1^	Found ± S.D. mg L^−1^	Spike Recovery (%)	Found ± S.D. mg L^−1^	
^a^ Treated wastewater	-	0	<LOD		<LOD	0.9
		0.25	0.22 ± 0.01	89	0.21 ± 0.014	
^a^ Mineral water	-	0	<LOD		<LOD	0.6
		0.25	0.24 ± 0.01	96	0.24 ± 0.015	
^b^ Cherry soft drink	10	0	2.95 ± 0.02	104	2.93 ± 0.02	0.3
		1	3.99 ± 0.07			
^b^ Strawberry drink	20	0	3.46 ± 0.02	101	3.44 ± 0.03	2.2
		1	4.47 ± 0.01			
^b^ Grape drink	10	0	1.72 ± 0.01	102	1.75 ± 0.04	0.1

^a^ sample containing norfloxacin, ^b^ samples containing allura red, ^c^ volume/volume sample; critical t-value: 2.91; LOD = detection limit; S.D = standard deviation.

**Table 3 molecules-28-02465-t003:** Quinine concentrations obtained from fluorimetric measurements in food samples (n = 3).

Samples	Webcam System Found ± S.D. mg L^−1^	Spectrofluorometer Found ± S.D. mg L^−1^	*t*-Test
Tonic Schweppes	81.4 ± 1.4	80.4 ± 1.2	1.2
Tonic water 2	65.3 ± 0.5	67.9 ± 1.8	2.0
Nordic Mix	62.1 ± 0.8	63.3 ± 1.2	1.9

Critical *t*-value: 2.91.

**Table 4 molecules-28-02465-t004:** Comparison of the system developed with other analytical methods for the determination of norfloxacin, allura red, and quinine in environmental and food samples, respectively.

Analytical Technique	Compounds	Type of Sample	LOD (mg L^−1^)	Intra-Day Precision (RSD%)	Portability	Method Cost	Type Image Processing	Reference
Fluorescence spectrometry	Norfloxacin	Wastewater, Surface water, and drinking water	0.022	NR	No	High	-	[62]
SPE-Spectrophotometric	Allura red	Tap water and wastewater	0.002	<7	No	High	-	[63]
Image digital colorimetry								
Smartphone detector	Norfloxacin	Pharmaceutical formulations	1.0	<5.9	Si	Low	Manual	[14]
SPE-Webcam detector	Norfloxacin	Tap, mineral, and treated wastewater	0.06	<3.7	Si	Low	Automatic	**This work**
Flatbed scanner	Allura Red	Beverages	0.60	<12%	No	Medium	Manual	[61]
Webcam detector	Allura red	Beverages	0.25	<1.3	Si	Low	Automatic	**This work**
Fluorescence digital image								
Smartphone	Quinine, rhodamine B, and riboflavin	Beverages	0.14	<7.3	Si	Low	Manual	[49]
Webcam detector	Quinine	Beverages	0.002	<2.6	Si	Low	Automatic	**This work**

SPE-Spectrophotometric: Solid phase extraction with spectrophotometric determination; NR: not reported. The limits of detection (LOD) and quantification (LOQ) were calculated as the concentration corresponding to three times and ten times the standard deviation of the blank solution (n = 10) over the calibration slope, respectively.

## Data Availability

Data available within the article or its Appendix A.

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
