# Peer review of "Multifunctional Portable System Based on Digital Images for In-Situ Detecting of Environmental and Food Samples"

_molecules, 2023, doi:10.3390/molecules28062465_

Round 1

Reviewer 1 Report

Authors and co-workers' paper entitled Multifunctional Portable System Based on Digital Images for In-situ Detecting Of Environmental And Food Samples" reported a method of developing a portable device created by 3D printing for colorimetric and fluorometric measurements, and the proposed device was tested by sensing norfloxacin, Allura red and quinine in the real sample. The resulting data were analyzed by t-test, proving the proposed devices were reliable. Although the manuscript focused on the utility technology but not the methodology, this work is nicely done, and the test data was entirely positive. The manuscript should be minor revised and respond to the following questions before being published in the Journal "Molecule."

The subsequent minor revisions are addressed:

1.      There are some format errors in the manuscript. Please check them very carefully before publication. For example, in Line 146, the "D" holder should be correct to the "3D" holder.

2.      The data in Figures S7 and S8 were not clear, and the photo in Figure 2 is also low resolution.  

3.      Figures 3 and 4 show that each calibration curve should display the independent analysis's relative standard deviation (RSD).

4.      In Figure 4, the analysis of the standard solution of norfloxacin (A, B) and Allura red (C, D) should provide the blank value (0 mg L-1)

5.      What is new in this manuscript? New process method for the 3D printing for colorimetric and fluorometric measurement devices or what?

6.      Please explain how to get the LOD and LOQ values in Table 1. How to determine these values?

7.      To detect the real sample, the authors should notice the matrix effect. The authors should illustrate the process to sense the real samples in the Experimental section. 

Author Response

Manuscript ID: molecules-2217109

Title: Multifunctional Portable System Based on Digital Images for In-Situ Detecting of Environmental and Food Samples

Authors: Víctor Cerdà Martín *, Diego Barzallo, Jorge Benavides, Edwin Palacio

Received: 29 January 2023

Reviewer 1

Authors and co-workers' paper entitled Multifunctional Portable System Based on Digital Images for In-situ Detecting of Environmental and Food Samples" reported a method of developing a portable device created by 3D printing for colorimetric and fluorometric measurements, and the proposed device was tested by sensing norfloxacin, Allura red and quinine in the real sample. The resulting data were analyzed by t-test, proving the proposed devices were reliable. Although the manuscript focused on the utility technology but not the methodology, this work is nicely done, and the test data was entirely positive. The manuscript should be minor revised and respond to the following questions before being published in the Journal "Molecule."

The subsequent minor revisions are addressed:

  1. There are some format errors in the manuscript. Please check them very carefully before publication. For example, in Line 146, the "D" holder should be correct to the "3D" holder.

Thank you for this observation. This line has been modified.

  1. The data in Figures S7 and S8 were not clear, and the photo in Figure 2 is also low resolution.

This has been modified

  1. Figures 3 and 4 show that each calibration curve should display the independent analysis's relative standard deviation (RSD).

This has been modified.

  1. In Figure 4 (Now 5), the analysis of the standard solution of norfloxacin (A, B) and Allura red (C, D) should provide the blank value (0 mg L-1)

This has been included

  1. What is new in this manuscript? New process method for the 3D printing for colorimetric and fluorometric measurement devices or what?

Thank you for raising this point. In other works, reported in literature, the images have been captured using the YouCam program and the deconvolution of the colors was carried out by ImageJ to obtain RGB data, which requires a further treatment for the interpretation of the results, i.e. conversion of RGB data to absorbance values, construction of calibration curves and linear regression fit, which requires more analysis time and experience in the treatment of digital images.

Thus, we have modified the sentence on page 12, line 411, as follows:

“The novelty of this 3D printed system with respect to those mentioned above and other colorimetric systems that have also been reported in the literature [14,45,65], are the versatility to performed both colorimetric and fluorimetric measurements simultaneously from processing of digital images in real time through the software developed in Matlab that includes various advantages: i) choice of the analytical technique to be used, ii) capture of the digital images, iii) construction of the calibration curves and iv) rapid acquisition of RGB channel results”.

  1. Please explain how to get the LOD and LOQ values in Table 1. How to determine these values?

The limits of detection (LOD) and quantification (LOQ) were calculated as the concentration corresponding to three times and ten times the standard deviation of the blank solution (n=10) over calibration slope, respectively.

  1. To detect the real sample, the authors should notice the matrix effect. The authors should illustrate the process to sense the real samples in the Experimental section.

As shown in Table 2, to assess matrix effects, environmental and food samples were analyzed without and with spiked with a known concentration. Recovery percentages obtained were from 89-102%, which indicates no significant matrix effects in quantitative analysis.

We have included a schematic representation about norfloxacin preconcentration procedure for further colorimetric analysis based on digital images as shown in Fig. 1 on page 4, line 144. However, allura red and quinine samples were directly analyzed, which only were diluted in a volumetric flask with deionized water and a 0.05 mol L-1 sulfuric acid solution respectively, as described on page 3, line 122.

Fig. 1. shows a schematic representation of the norfloxacin preconcentration procedure for further colorimetric analysis based on digital images.

Reviewer 2 Report

The manuscript describes a 3D-printed device equipped with a web camera and LED light sources to detect norfloxacin, Allura red, and quinine in water and beverage samples. The topic of the portable optic detection system is interesting; the results from the proposed device and conventional spectrophotometer/spectrofluorometer are comparable. Other comments:

·       The introduction is insufficient; several original and classical works using LED or similar light sources have not been discussed in the current manuscript, such as Lab on a Chip 9 (5), 733 and Lab on a Chip 5 (10), 1041.

·       The design criteria for the optical system are unclear, which may influence the detection performance of the fluorescent signal. It seems that the author used monochromatic LED and the inherent filters of the RGB detectors in the web camera. However, the monochromatic LED emits a broad range of wavelengths rather than a single wavelength emission by a laser diode. The wavelength band for the RGB detector may not be sufficient to block the light from the LED. Thus, it is surprising that the detection limit of the portable system is comparable to the benchtop spectrofluorometer. The authors should provide a detailed discussion of this aspect.

·       The font of sections is not consistent; e.g., the 3.2 section is in italics. The alignment in Table 2 is not consistent. The title of section 3.1.1 is incomplete. 

Round 2

Reviewer 2 Report

The authors have sufficiently addressed the concerns. I thus recommend the acceptance of the revised manuscript.